# Psychometric Validation of the German Translation of the Quality of Life Questionnaire-Bronchiectasis (QOL-B)—Data from the German Bronchiectasis Registry PROGNOSIS

**DOI:** 10.3390/jcm11020441

**Published:** 2022-01-15

**Authors:** Laura Quellhorst, Grit Barten-Neiner, Andrés de Roux, Roland Diel, Pontus Mertsch, Isabell Pink, Jessica Rademacher, Sivagurunathan Sutharsan, Tobias Welte, Annegret Zurawski, Felix C. Ringshausen

**Affiliations:** 1Department of Respiratory Medicine, Hannover Medical School (MHH), 30625 Hannover, Germany; LauraQuellhorst@gmx.de (L.Q.); Pink.Isabell@mh-hannover.de (I.P.); Rademacher.Jessica@mh-hannover.de (J.R.); Welte.Tobias@mh-hannover.de (T.W.); Zurawski.Annegret@mh-hannover.de (A.Z.); 2CAPNETZ STIFTUNG, Hannover Medical School (MHH), 30625 Hannover, Germany; Barten.Grit@mh-hannover.de; 3Biomedical Research in End-Stage and Obstructive Lung Disease Hannover (BREATH), German Center for Lung Research (DZL), 30625 Hannover, Germany; 4Pneumologische Praxis am Schloss Charlottenburg, 14059 Berlin, Germany; drderoux.a@gmail.com; 5Institute for Epidemiology, University Medical Center Schleswig-Holstein, 24105 Kiel, Germany; roland.diel@epi.uni-kiel.de; 6LungenClinic Grosshansdorf, Airway Research Center North (ARCN), German Center for Lung Research (DZL), 22927 Grosshansdorf, Germany; 7Institution for Statutory Accident Insurance and Prevention in the Health and Welfare Services (BGW), 22089 Hamburg, Germany; 8Department of Medicine V, University Hospital, LMU Munich, Comprehensive Pneumology Center Munich (CPC-M), German Center for Lung Research (DZL), 81377 Munich, Germany; Pontus.Mertsch@med.uni-muenchen.de; 9Department of Pulmonary Medicine, University Hospital Essen, Ruhrlandklinik, University Duisburg-Essen, 45239 Essen, Germany; Sivagurunathan.Sutharsan@rlk.uk-essen.de

**Keywords:** bronchiectasis, Germany, patient-reported outcome measures, quality of life, questionnaire design, registries

## Abstract

Patients with bronchiectasis feature considerable symptom burden and reduced health-related quality of life (QOL). We provide the psychometric validation of the German translation of the disease-specific Quality of Life Questionnaire-Bronchiectasis (QOL-B), version 3.1, using baseline data of adults consecutively enrolled into the prospective German bronchiectasis registry PROGNOSIS. Overall, 904 patients with evaluable QOL-B scores were included. We observed no relevant floor or ceiling effects. Internal consistency was good to excellent (Cronbach’s α ≥0.73 for each scale). QOL-B scales discriminated between patients based on prior pulmonary exacerbations and hospitalizations, breathlessness, bronchiectasis severity index, lung function, sputum volume, *Pseudomonas aeruginosa* status and the need for regular pharmacotherapy, except for Social Functioning, Vitality and Emotional Functioning scales. We observed moderate to strong convergence between several measures of disease severity and QOL-B scales, except for Social and Emotional Functioning. Two-week test-retest reliability was good, with intraclass correlation coefficients ≥0.84 for each scale. Minimal clinical important difference ranged between 8.5 for the Respiratory Symptoms and 14.1 points for the Social Functioning scale. Overall, the German translation of the QOL-B, version 3.1, has good validity and test-retest reliability among a nationally representative adult bronchiectasis cohort. However, responsiveness of QOL-B scales require further investigation during registry follow-up.

## 1. Introduction

Bronchiectasis is a chronic suppurative and often progressive airway disease, which manifests with considerable symptom burden, in particular persistent productive cough, reduced exercise capacity, frequent pulmonary exacerbations and reduced health-related quality of life (QOL) [1]. Its prevalence has been increasing over the past years in many settings, resulting in substantial economic burden to healthcare systems [2,3,4].

In randomized controlled trials (RCTs) in bronchiectasis, various objective measures such as sputum bacterial load or pulmonary exacerbations, but also patient-reported outcome measures (PROMs) such as QOL have been used to evaluate the efficacy of different therapeutic interventions [5,6,7,8]. In this regard, PROMs are not only important as a patient priority [9], but have gained importance for regulatory agencies as trial endpoints [10,11].

So far, a variety of different tools have been applied to assess QOL in bronchiectasis populations, either using well-established respiratory health-related QOL questionnaires such as the St. George’s Respiratory Questionnaire (SGRQ) or disease-specific QOL questionnaires such as the Quality of Life Questionnaire-Bronchiectasis (QOL-B), which is the first tool specifically developed for use in bronchiectasis [8,12,13,14,15]. To validate those questionnaires accepted measures of symptom burden and disease severity have been used, including breathlessness, sputum volume, forced expiratory volume in one second (FEV_1_), prior pulmonary exacerbations and hospitalizations, infection status and radiology [16,17].

While the disease-specific QOL-B has been validated in an international multi-center RCT as well as a Spanish bronchiectasis population [12,18], its correlates with symptom burden and disease severity among German adults with bronchiectasis are unknown.

In order to investigate QOL among this population, the Prospective German Non-CF Bronchiectasis Patient Registry (PROGNOSIS) applies the respective translation of the QOL-B, version 3.1, at baseline visits and during annual follow-up [19], thus offering an opportunity to explore its validity in a real-world application. Therefore, the aim of our study was to provide a psychometric validation of the German translation of the QOL-B, version 3.1, using baseline data of the first 1000 patients consecutively enrolled into PROGNOSIS.

## 2. Materials and Methods

### 2.1. Database

In summary, PROGNOSIS (www.bronchiektasen-register.de, accessed on 12 January 2022; registered at Clinicaltrials.gov under the identifier NCT02574143; hosted at Hannover Medical School [MHH], Hannover, Germany) is a currently ongoing, prospective and non-interventional registry, which was launched in June 2015 with the initial aim to recruit 750 patients with computed tomography (CT)-confirmed bronchiectasis from at least 10 sites across Germany over 3 consecutive years [19]. Since then, >1500 patients have been enrolled from a total of 38 sites across all levels of healthcare, including 15 respiratory physicians in private practice as well as 13 teaching and 10 university hospitals due to expanded funding, an associated partnership with the German Center for Lung Research (DZL, Giessen, Germany) and the continuous support of CAPNETZ STIFTUNG (www.capnetz.de, accessed 12 January 2022). This ensured broad geographic and epidemiological representativeness, with about half of patients enrolled at university hospitals and one quarter at private practices and teaching hospitals, each. PROGNOSIS prospectively collects baseline, annual (±3 months) follow-up and outcome data, using a standardized electronic case report form (eCRF) via an online database accessible through the registry’s homepage. Pseudonymization and electronic storage of clinical data follow a uniform and quality-assured standard operating procedure according to German and European Data Protection Act. The PROGNOSIS protocol and eCRF have been harmonized with those of the European bronchiectasis registry EMBARC (www.bronchiectasis.eu, accessed 12 January 2022) in order to allow data sharing and joint data analyses. We recorded comprehensive clinical data including not only demographics, disease history, etiological testing, complications and comorbidities, microbiology, pulmonary function, radiology and treatment, but also measures of disease severity and symptom burden. The multidimensional Bronchiectasis Severity Index (BSI) was calculated as previously described [17]. Pulmonary exacerbations were defined as the need for a significant change in medical management for acute respiratory symptoms, typically requiring antibiotics, and recorded from either patient history, hospital and/or prescription records or a combination of those. Etiology was recorded by the site investigators and verified centrally using the patients’ etiological testing data. Chronic infection was defined as two isolates of the same pathogen at least 3 months apart over 1 year while in a stable state [1]. Spirometry was performed according to American Thoracic Society/European Respiratory Society standards and predicted values were calculated centrally using Global Lung Function Initiative equations. Radiological severity was assessed locally by the site investigators in the patients’ most recent CT scans according to the modified Reiff score [20]. All datasets were externally validated and, thereafter, manually cleaned by the authors before analysis. PROGNOSIS received ethical approval by the ethic committees of all participating centers, referring to the initial ethical approval of MHH’s institutional review board (No. 6656/2015).

### 2.2. Patient Population and Study Design

Inclusion criteria are age ≥18 years, CT-confirmed bronchiectasis and prior written and informed consent, while known cystic fibrosis at the time of inclusion and prior lung/heart-lung transplantation exclude patients from participation. For the present analysis, only patients were selected in whom at least one baseline QOL-B scale was evaluable based on responses to the respective items. We conducted a cross-sectional analysis of baseline data with the aim to provide the psychometric validation of the German translation of the QOL-B, version 3.1, though excluding the evaluation of responsiveness over time.

### 2.3. Quality of Life Assessment

Disease-specific QOL was determined at the baseline visit by means of the German translation of the QOL-B, version 3.1, for which prior written permission of the copyright holder had been obtained [12,21]. However, completion of QOL-B was optional. Patients were instructed to answer the QOL-B on their own (self-administered), while at the hospital or during a clinic or private practice visit. The QOL-B consists of 37 items that subdivide into 8 scales (Respiratory Symptoms, Physical Functioning, Vitality, Role Functioning, Health Perceptions, Emotional Functioning, Social Functioning and Treatment Burden), with scores ranging from 0 to 100 within each scale. No total score is calculated. Patients not receiving bronchiectasis treatment were instructed to skip the Treatment Burden scale. All items were scored using the *SAS and SPSS Program Codes for Scoring the QOL-B Version 3.1*, according to the copyright holder’s instructions, and rechecked manually by random sampling (Appendix A) [12,21]. If responses were missing for more than half the items in a scale, the score for that scale was not calculated. Missing QOL-B values were not imputed (Appendix A). Discriminant and convergent validity of QOL-B scales were assessed using pulmonary exacerbations, hospitalizations, breathlessness (Medical Research Council [MRC] dyspnea scale), BSI, FEV_1_, patient-estimated average daily sputum volume, *Pseudomonas aeruginosa* infection status, need for regular pharmacological treatment, radiological severity and prior thoracic surgery as markers of bronchiectasis severity and symptom burden.

### 2.4. Statistical Analysis

Continuous data are presented as mean with standard deviation (SD) or median with interquartile range according to their distribution, categorical data as numbers and percentages. The Kolmogorov–Smirnov test was used to assess distribution of continuous data. Floor and ceiling effects were assessed by descriptive statistics. We assumed floor and ceiling effects to be relevant, if ≥15% of respondents had scores of 0 and 100 within a scale, respectively [12,16]. Internal consistency was evaluated using Cronbach’s α. Short-term test–retest reliability (reproducibility) was estimated with intraclass correlation coefficients (ICCs) in a subgroup of 20 randomly selected patients at MHH’s adult bronchiectasis clinic in the absence of a change in clinical status over a 14(±7)-day period. The Mann–Whitney U and the Kruskal–Wallis test were used to assess differences between groups in the analysis of discriminant validity, while Spearman’s correlations were calculated to assess convergent validity of QOL-B scales. Rho (*r*) values <0.3 indicate weak, values of 0.3 to 0.49 moderate and values >0.5 strong correlation [15,22]. We considered a two-sided *p*-value < 0.05 statistically significant. Minimal clinical important difference (MCID) estimates were assessed distribution-based by the standard error of the mean method (SEM = SD √(1-α)). Overall, missing values were infrequently observed in spirometry (FEV_1_ 6.4%), breathlessness (MRC dyspnea scale 5.8%), number of pulmonary exacerbations in the previous 12 months (2.6%) and body mass index (0.2%). In order to utilize all datasets with evaluable QOL-B scales we calculated missing values by the multiple imputation method, with age, sex, comorbidities, etiology, prior hospitalizations, *Pseudomonas aeruginosa* infection status and radiological severity as predictor variables. The number of imputations was set to 10. All analyses were performed with SPSS, version 27.0 (IBM Corp., Armonk, NY, USA).

## 3. Results

We consecutively recruited 1000 adults with CT-confirmed bronchiectasis into the PROGNOSIS registry from July 2015 to March 2018. Of those, 904 patients (90.4%) completed the QOL-B at baseline and had at least one evaluable baseline QOL-B scale, representing the final study population for the present analysis. Table 1 shows the demographic and baseline characteristics, which were comparable between patients who had filled out the QOL-B and those who had not (Appendix A).

Patients had a mean (SD) age of 59.5 (15.8) years, ranging from 18 to 93 years, showed a slight female predominance (59.5%) and, on average, had moderate airflow obstruction on spirometry (mean [SD] FEV_1_ 69.8% [26.8] predicted; mean [SD] Tiffeneau index 0.69 [0.15]; Table 1). The five most common etiologies of bronchiectasis were idiopathic in 337 (37.3%), postinfectious in 180 (19.9%), COPD in 133 (14.7%), asthma in 99 (11.0%) and primary ciliary dyskinesia/Kartagener syndrome in 79 patients (8.7%). Likewise, the most common comorbidities with a frequency of >10% were cardiovascular in 354 (39.2%), chronic rhinosinusitis in 270 (29.9%), asthma and COPD in each 269 (29.8%), gastro-esophageal reflux in 166 (18.4%), nasal polyps in 141 (15.6%), malignancy in 108 (11.9%) and osteoporosis in 98 (10.8%). Depression and anxiety disorders were reported in 86 (9.5%) and 34 patients (3.8%), respectively (Appendix A). *Pseudomonas aeruginosa* was the most common respiratory pathogen in 223 of 680 patients (32.8%), of whom sputum culture results were available at baseline, followed by *Staphylococcus aureus* in 112 (16.5%), *Haemophilus influenzae* in 93 (13.7%), *Aspergillus fumigatus* in 73 (10.7%) and nontuberculous mycobacteria in 41 patients (6.0%; Appendix A).

### 3.1. Floor and Ceiling Effects, Internal Consistency and Test-Retest Reliability

Mean baseline QOL-B scores, the assessment of floor (score = 0) and ceiling effects (score = 100) as well as internal consistency are shown in Table 2. In summary, we observed no floor and ceiling effects in ≥15% of patients. Although 10.7% of patients had a Physical Functioning score of 0, we found that ≤4.7% had a score of 0 on any other scale. Similarly, 10.9%, 9.1% and 7.0% of patients showed an Emotional, Social and Role Functioning score of 100, respectively, but ≤4.7% had a score of 100 on any other scale (Table 2). Moreover, we evaluated internal consistency using Cronbach’s α. We found that all values were ≥0.73, thus indicating a coherent and reliable construct with strong correlation among the items on each scale (Table 2). Next, we evaluated test-retest reliability in a subgroup of 20 randomly selected subjects, who repeated the QOL-B in the absence of a change in clinical status within a 14(±7)-day interval. Here, we observed ICCs ≥0.84 for each scale, indicating good to excellent reproducibility (Table 2).

### 3.2. Discriminant and Convergent Validity of QOL-B Scores

In order to assess discriminant validity, we compared mean baseline QOL-B scores with a variety of measures of disease severity and symptom burden. We found significant discrimination between subjects by means of pulmonary exacerbations (Table 3) and hospitalizations (Table 4) as well as MRC dyspnea scale (Figure 1; Appendix A) for all QOL-B scales (*p* < 0.001 for each scale).

All QOL-B scales were worse in patients with more severe bronchiectasis according to BSI categories (Figure 2; Appendix A). With regard to FEV_1_, QOL-B scales discriminated patients (*p* < 0.01 for each scale), except Social Functioning (Figure 3; Appendix A). In addition, QoL-B scales were strongly associated with categories of average sputum volume, *Pseudomonas aeruginosa* infection status and regular pharmacological treatment of bronchiectasis (*p* < 0.01 for each scale), with the exception of Vitality and Emotional Functioning (Figure 4; Appendix A; Table 5; Appendix A).

In contrast, only the QOL-B scales Physical Functioning (*p* = 0.022), Role Functioning (*p* = 0.027) and Health Perceptions (*p* = 0.048) were significantly worse in patients with ≥3 lobes affected and/or cystic bronchiectasis, whereas only the Respiratory Symptoms and Social Functioning scales were significantly worse in patients with prior thoracic surgery (*p* = 0.025 and *p* = 0.002, respectively; Appendix A).

Moreover, we observed moderate to strong convergence between breathlessness and Respiratory Symptoms, Physical Functioning, Vitality, Role Functioning, Health Perceptions and Treatment Burden (*r* = −0.315 to −0.620; *p* < 0.001, each; Table 6). Moderate correlations were found for (a) FEV_1_ and Respiratory Symptoms, Physical Functioning and Role Functioning (*r* = 0.304 to 0.470; *p* < 0.001, each); (b) pulmonary exacerbation rate and Role Functioning (*r* = −0.305, *p* < 0.001*)*; (c) hospitalization rate and Role Functioning (*r* = −0.339, *p* < 0.001); as well as (d) average daily sputum volume and Respiratory Symptoms (*r* = −0.321, *p* <0.001; Table 6).

### 3.3. Minimal Clinical Important Difference

The derived MCIDs ranged between 8.5 points for the Respiratory Symptoms scale and 14.1 points for the Social Functioning scale (Table 7).

## 4. Discussion

The present study confirms that the German translation of the QOL-B, version 3.1, is a valid tool for the assessment of disease-specific QOL among adult patients with bronchiectasis. We showed that it is internally consistent, reproducible over a 2-week period and has good construct validity, with most scales discriminating between patients based on symptom burden as well as established measures of disease severity of bronchiectasis.

Our findings are comparable to those of the original QOL-B validation study by Quittner and colleagues [12] as well as a validation study of the QOL-B, version 3.0, by Olveira and colleagues [18]. Overall, we observed only small floor or ceiling effects, which is a prerequisite for the detection of future changes [10]. While slightly more patients had a QOL-B score of “0” compared to the studies by Quittner et al. and Olveira et al. (maximum 10.7% vs. 5.1% and 3.4%, respectively), scores of “100” were less frequently observed in our study (maximum 10.9% vs. 24.1% and 21.7%, respectively) [12,18]. We demonstrated good to excellent internal consistency, with Cronbach’s α ranging from 0.73 to 0.92, which is accordance with the results of the two previous studies (Cronbach’s α 0.70–0.91, each) [12,18]. Similarly, we found good to excellent test-retest reliability in line with the above-mentioned studies (ICCs 0.84–0.96 vs. 0.72–0.86 and 0.68–0.88, respectively) [12,18], thus supporting the appropriateness of our reproducibility analysis, even if clearly less datasets were available for assessment. Moreover, our MCID estimates of QOL-B scales showed a range of 8.5–14.1 points, with 8.5 points for the Respiratory Symptoms scale. This is in agreement with the MCID ranges of 7.7–12.6 and 8.2–13.3 points, with a MCID of 7.7 and 8.2 points for the Respiratory Symptoms scale in the AIR-BX1 and AIR-BX2 studies, respectively [12], when determined by the SEM method. Similarly, in the Spanish validation study the MCID was 6.8 points for the Respiratory Symptoms scale [18]. In order to establish discriminant and convergent validity we analyzed several established measures of health status and disease severity and found good discrimination as well as moderate to strong correlations with the majority of the QOL-B scales, with the exception of the Vitality, Emotional Functioning and Social Functioning scales.

However, we also found some differences when comparing our results to those of previous studies on the QOL-B’s validity. In our study, all QOL-B scales except Social Functioning discriminated very well between patients based on FEV_1_, with moderate correlations found between Respiratory Symptoms, Physical Functioning as well as Role Functioning scales and FEV_1_. In this regard, the respective results by Quittner et al. were less strong in the setting of two RCTs, with merely moderate correlation found between Physical Functioning and FEV_1_ in the AIR-BX2 trial [12]. Furthermore, we observed that the Treatment Burden scale was generally worse in patients with more severe bronchiectasis and higher symptom burden. This is in contrast to the study by Olveira et al., who found that the Treatment Burden scale discriminated only between patients according to radiological severity, as assessed by the Bhalla score [18]. In this regard, one should keep in mind that in our study 259 subjects (28.7%) skipped the Treatment Burden scale, as they were not receiving bronchiectasis treatment, potentially limiting its relevance to the general bronchiectasis population.

The existing QOL questionnaires, which have been used in bronchiectasis studies, have advantages and disadvantages. The SGRQ has been most widely applied. While it has good psychometric properties, it consists of 50 items, thus making it rather complex [8,16]. Moreover, it has primarily been developed for use in other chronic respiratory conditions, with a greater focus on breathlessness, and has never obtained broad acceptance for routine use in Germany. In this regard, the lower respiratory tract infections—visual analogue scale (LRTI-VAS) [23], the COPD Assessment Test (CAT) [24] and the Leicester Cough Questionnaire (LCQ) [25] are more feasible, but are neither specific for use in bronchiectasis, with the later mainly focusing on health related to cough.

The QOL-B was the first questionnaire specifically developed for the assessment of QOL among patients with bronchiectasis [12,21]. It has 37 items resulting in 8 scales, with no overall score provided. Data on the responsiveness of the Respiratory Symptoms scale from RCTs are conflicting [16]. While the QOL-B Respiratory Symptoms scale significantly improved along with reductions of bacterial load during on-treatment periods in the RCTs ORBIT-3 and -4 [26], it turned out to be unresponsive in several other RCTs despite evidence for other clinical meaningful changes [27,28,29]. However, this observation has previously been made for the SGRQ and the LCQ [29,30,31], indicating that, amongst others, responsiveness may depend on differences in patient populations and the type of intervention [14,16].

Therefore, other disease-specific QOL questionnaires have been proposed as alternatives to the QOL-B [14,15]. The Bronchiectasis Health Questionnaire (BHQ) is brief and simple-to-use. Its 10 items result in a single, one-dimensional overall score [14]. While psychometric data supporting its use are available, its responsiveness and MCID have not been studied in detail yet [16]. Very recently, the Bronchiectasis Impact Measure (BIM) has been developed and validated as a simple PROM [15]. In contrast to other questionnaires, it is composed of 8 visual analogue scales representing unique domains and focuses on the impact of bronchiectasis symptoms on patients’ lives. This appears to be particularly useful with ambiguous symptoms, which are typically perceived by patients with considerable interindividual variability, such as sputum, with some patients regarding increased sputum production as positive, while others think of it as negative [8]. Notably, this allowed the authors of the BIM to provide patient-derived MCID estimates [15]. While the BIM was strongly to very strongly correlated with QOL-B scores, it appeared to outperform the QOL-B by means of convergent validity (and the degree of floor and ceiling effects), with most of its scales showing moderate to very strong correlation with measures of disease severity [15]. However, direct comparison between both measures of disease-specific QOL from multicenter studies is lacking. A recent comprehensive systematic review and meta-analysis of health-related QOL questionnaires in bronchiectasis concludes that good psychometric data on the bronchiectasis-specific QOL questionnaires are emerging, but further studies on their medium- to long-term test-retest reliability, responsiveness and MCID are required. Currently, only the CAT, the QOL-B and the BIM have provided MCID values for bronchiectasis populations [16].

Our study has strengths and limitations. Most importantly, the data underlying our study were captured at 38 sites from all levels of the German healthcare system with the broadest possible geographical distribution, including not only secondary and tertiary care centers but also practitioners in private practice, and may therefore be regarded both nationally representative and related to real life. As a consequence, we were able to include a large number of patients as well as a multitude of measure of health status and disease severity in our analysis. However, the key limitation is that we were unable to evaluate responsiveness due to our cross-sectional study design. In this regard, we will utilize the registry data as fully validated follow-up datasets will become available in the near future. Another limitation is that we used a distribution-based method of estimating MCIDs only (SEM method). In contrast to anchor-based or patient-derived MCIDs, which may be more clinically relevant as they evaluate the direct impact on the patients, our MCIDs are mathematically derived and essentially measure the variances of scores [15]. However, there is no single agreed method of estimating the MCID for PROMs and commonly accepted anchors are unavailable for most of the bronchiectasis-associated symptoms [15,32]. Lastly, we decided not to include additional questionnaires as further PROMs for comparison to QOL-B scores, as provided by other studies [12,15,18,23], considering feasibility of data collection within our registry. Therefore, we are unable to draw conclusions about the superiority of a particular tool for the assessment of QOL in patients with bronchiectasis in our setting.

Even though our findings may be largely confirmative, our study is of importance as until now there was no validated tool ready for use among the German bronchiectasis population. The epidemiology of bronchiectasis and the standard of care, keeping the absence of any licensed pharmacological treatment in mind, show distinct differences between geographical regions and healthcare systems, thus supporting the need for studies in different bronchiectasis populations and justifying the present analysis [33]. QOL is not only an important PROM in RCTs but also matters to patients in their everyday lives [9]. Our findings provide evidence that the QOL-B is a valid tool and offer guidance for its interpretation in clinical practice. In this respect, our study may contribute to the recognition of disease-specific QOL as a crucial criterion for clinical decision-making in the routine management of bronchiectasis in Germany.

## 5. Conclusions

We validated the German translation of the QOL-B, version 3.1, in a representative cohort of adult German patients with bronchiectasis. However, its responsiveness needs to be established during registry follow-up by means of changes in health status, including clinical deterioration due to pulmonary exacerbations as well as improvement due to continuous specialized care and targeted therapeutic interventions.

## Figures and Tables

**Figure 1 jcm-11-00441-f001:**
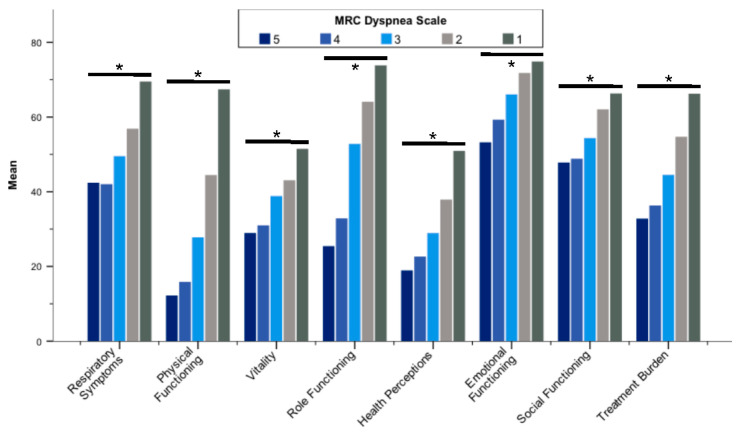
Mean Quality of Life Questionnaire-Bronchiectasis scores stratified by the MRC dyspnea scale. Abbreviation: MRC, Medical Research Council. * *p* < 0.001. Differences between groups were assessed by the Kruskal–Wallis test.

**Figure 2 jcm-11-00441-f002:**
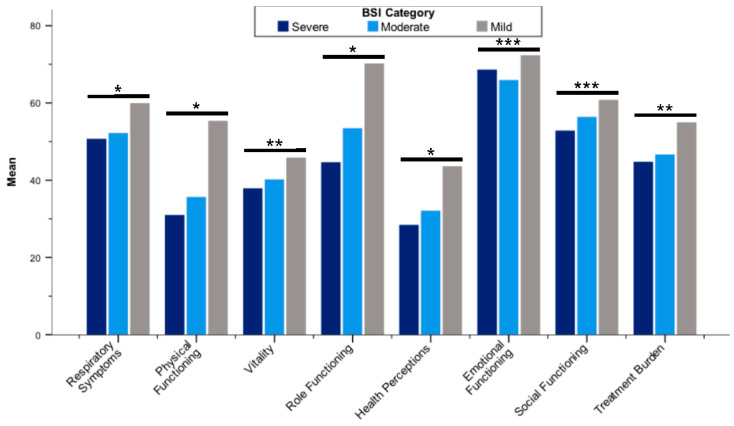
Mean Quality of Life Questionnaire-Bronchiectasis scores stratified by BSI categories. Abbreviation: BSI, Bronchiectasis Severity Index; * *p* < 0.001; ** *p* < 0.01; *** *p* < 0.05. Differences between groups were assessed by the Kruskal–Wallis test.

**Figure 3 jcm-11-00441-f003:**
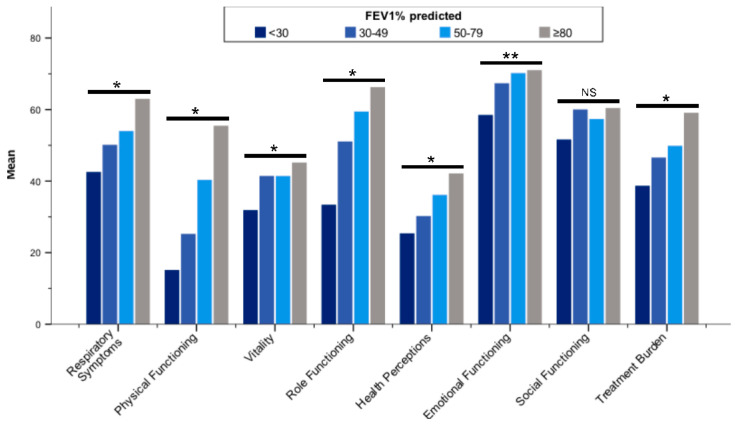
Mean Quality of Life Questionnaire-Bronchiectasis scores stratified by ppFEV_1_ (categorized). Abbreviations: ppFEV_1_, forced expiratory volume in one second (% predicted); NS, not significant. * *p* < 0.001; ** *p* < 0.01. Differences between groups were assessed by the Kruskal–Wallis test.

**Figure 4 jcm-11-00441-f004:**
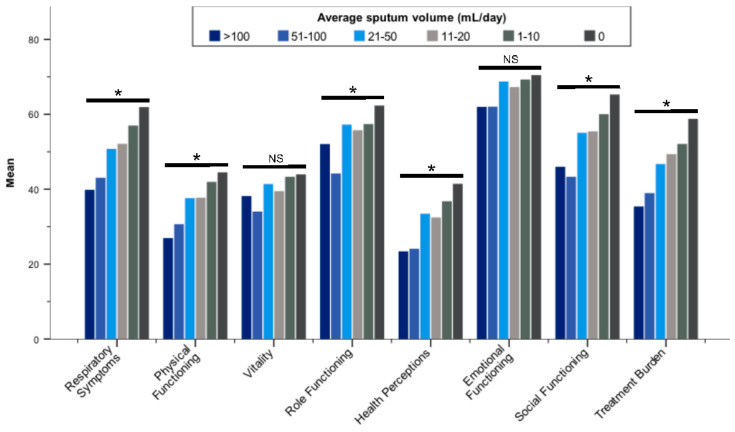
Mean Quality of Life-Bronchiectasis scores stratified by average daily sputum volume (categorized). Abbreviation: NS, not significant. * *p* ≤ 0.001; Differences between groups were assessed by the Kruskal–Wallis test.

**Table 1 jcm-11-00441-t001:** Demographic and baseline characteristics (*n* = 904).

Variable	Value
Age (years), mean (SD)	59.5 (15.8)
Females, n (%)	538 (59.5)
BMI (kg/m^2^), mean (SD)	24.1 (4.5)
FEV_1_ %predicted, mean (SD)	69.8 (26.8)
≥80%predicted, n (%)	344 (38.1)
50–79%predicted, n (%)	322 (35.6)
30–49%predicted, n (%)	180 (19.9)
<30%predicted, n (%)	58 (6.4)
Radiological severity, n (%)	
<3 lobes affected	325 (36.0)
≥3 lobes affected and/or cystic bronchiectasis	579 (64.0)
MRC dyspnea scale, n (%)	
1–3	751 (83.1)
4–5	153 (16.9)
Smoking, n (%)	
Active smoker	57 (6.3)
Former smoker	335 (37.1)
Never smoked	512 (56.6)
Exacerbations in the past 12 months, median (IQR)	1 (0–3)
0, n (%)	271 (30.0)
1–2, n (%)	361 (39.9)
≥3, n (%)	272 (30.1)
Prior hospital admission, n (%) ^1^	349 (38.6)
Hospitalizations in the past 12 months, median (IQR) ^1^	0 (0–1)
Regular pharmacological treatment of bronchiectasis, n (%)	704 (77.9)
Regular sputum production, n (%)	704 (77.9)
Average daily sputum volume, median (IQR)	20 (10–50)
0 mL/day, n (%)	284 (31.4)
1–10 mL/day, n (%)	261 (28.9)
11–20 mL/day, n (%)	122 (13.5)
21–50 mL/day, n (%)	151 (16.7)
51–100 mL/day, n (%)	60 (6.6)
>100 mL/day, n (%)	26 (2.9)
BSI category (*n* = 666), n (%) ^2^	
Mild (0–4)	150 (22.5)
Moderate (5–8)	390 (58.6)
Severe (≥9)	126 (18.9)

^1^ Hospitalization due to severe pulmonary exacerbation. ^2^ We could not calculate BSI in all patients due to missing data on repeat sputum microbiology defining chronic infection. Abbreviations: BMI, body mass index; BSI, Bronchiectasis Severity Index; COPD, chronic obstructive pulmonary disease; FEV_1_, forced expiratory volume in 1 s; IQR, interquartile range; MRC, Medical Research Council; QOL-B, QOL-B, Quality of Life Questionnaire-Bronchiectasis; SD, standard deviation.

**Table 2 jcm-11-00441-t002:** QOL-B scores at baseline, floor and ceiling effects, internal consistency and test-retest reliability.

QOL-B Scale	*n* ^1^	Mean (SD) QOL-B Scores	Floor Effects, n (%)	Ceiling Effects, n (%)	Cronbach’s α	ICC (95% CI)
Respiratory Symptoms	892	56.2 (21.0)	1 (0.1)	9 (1.0)	0.84	0.93 (0.82–0.97)
Physical Functioning	889	41.8 (29.8)	95 (10.7)	42 (4. 7)	0.92	0.96 (0.90–0.99)
Vitality	892	42.0 (21.4)	40 (4.5)	10 (1.1)	0.76	0.94 (0.84–0.97)
Role Functioning	898	58.8 (27.4)	23 (2.6)	63 (7.0)	0.86	0.88 (0.69–0.95)
Health Perceptions	891	36.3 (22.6)	42 (4.7)	3 (0.3)	0.79	0.84 (0.59–0.94)
Emotional Functioning	889	69.2 (21.9)	4 (0.4)	97 (10.9)	0.82	0.94 (0.85–0.98)
Social Functioning	878	59.9 (26.9)	31 (3.5)	80 (9.1)	0.73	0.87 (0.66–0.95)
Treatment Burden	645	51.3 (25.1)	18 (2.8)	27 (4.2)	0.73	0.90 (0.73–0.96)

^1^ If responses were missing for more than half the items in a scale, the score for that scale was not calculated. Patients not receiving bronchiectasis treatment were instructed to skip the Treatment Burden scale. Abbreviations: CI, confidence interval; ICC, intraclass correlation coefficient; QOL-B, Quality of Life Questionnaire-Bronchiectasis; SD, standard deviation.

**Table 3 jcm-11-00441-t003:** Discrimination of QOL-B scores, stratified by history of pulmonary exacerbations in the previous 12 months (categorized).

Mean (SD) QOL-B Scores at Baseline According to Pulmonary Exacerbations				
QOL-B Scale	0	1–2	≥3	*p*-Value ^1^
Respiratory Symptoms	62.0 (20.7)	56.8 (19.9)	49.7 (20.7)	<0.001
Physical Functioning	47.6 (30.5)	44.3 (30.1)	33.3 (26.5)	<0.001
Vitality	46.3 (21.8)	43.7 (20.9)	36.3 (20.1)	<0.001
Role Functioning	67.7 (25.0)	60.9 (26.1)	47.7 (27.4)	<0.001
Health Perceptions	42.9 (24.0)	37.9 (21.7)	27.9 (19.5)	<0.001
Emotional Functioning	73.4 (20.3)	71.1 (20.0)	63.0 (23.7)	<0.001
Social Functioning	68.5 (23.4)	60.2 (25.7)	51.2 (28.8)	<0.001
Treatment Burden	59.2 (24.1)	52.4 (24.7)	42.7 (24.0)	<0.001

^1^ Differences between groups were assessed by the Kruskal–Wallis test. Abbreviations: QOL-B, Quality of Life Questionnaire-Bronchiectasis; SD, standard deviation.

**Table 4 jcm-11-00441-t004:** Discrimination of QOL-B scores, stratified by history of prior hospitalization in the previous 12 months.

Mean (SD) QOL-B Scores at Baseline According to Prior Hospitalization			
QOL-B Scale	Yes	No	*p*-Value ^1^
Respiratory Symptoms	51.3 (21.3)	59.2 (20.3)	<0.001
Physical Functioning	32.4 (27.3)	47.7 (29.8)	<0.001
Vitality	39.0 (21.5)	43.9 (21.1)	<0.001
Role Functioning	48.4 (27.0)	65.4 (25.6)	<0.001
Health Perceptions	31.6 (21.1)	39.4 (22.9)	<0.001
Emotional Functioning	66.5 (22.9)	70.8 (21.1)	<0.001
Social Functioning	55.2 (27.8)	62.8 (25.9)	<0.001
Treatment Burden	44.5 (24.2)	55.9 (24.7)	<0.001

^1^ Differences between groups were assessed by the Mann–Whitney U test. Abbreviations: QOL-B, Quality of Life Questionnaire-Bronchiectasis; SD, standard deviation.

**Table 5 jcm-11-00441-t005:** Discrimination of QOL-B scores, stratified by *Pseudomonas aeruginosa* infection at baseline and/or in the previous 12 months.

Mean (SD) QOL-B Scores According to *Pseudomonas aeruginosa* Infection			
QOL-B Scale	Yes	No	*p*-Value ^1^
Respiratory Symptoms	51.0 (20.1)	59.0 (21.0)	<0.001
Physical Functioning	35.2 (28.2)	45.5 (30.0)	<0.001
Vitality	40.5 (21.4)	42.9 (21.3)	0.080
Role Functioning	52.3 (27.4)	62.4 (26.7)	<0.001
Health Perceptions	31.7 (20.3)	38.8 (23.4)	<0.001
Emotional Functioning	67.2 (22.5)	70.2 (21.5)	0.063
Social Functioning	54.5 (26.1)	62.8 (26.9)	<0.001
Treatment Burden	45.8 (23.6)	55.0 (25.5)	<0.001

^1^ Differences between groups were assessed by the Mann–Whitney U test. Abbreviations: QOL-B, Quality of Life Questionnaire-Bronchiectasis; SD, standard deviation.

**Table 6 jcm-11-00441-t006:** Convergent validity: correlation between baseline QOL-B scores and measures of health status.

Correlations of Baseline Scores on QOL-B Scales with Measures of Health Status					
QOL-B Scale	ppFEV_1_	Pulmonary Exacerbation Rate	Hospitalization Rate	MRC Dyspnea Scale	Average Sputum Volume (mL/Day)
Respiratory Symptoms (*n* = 892)	*r* = 0.304 *p* < 0.001	*r* = −0.241 *p* < 0.001	*r* = −0.223 *p* < 0.001	*r* = −0.414 *p* < 0.001	*r* = −0.321 *p* < 0.001
Physical Functioning (*n* = 889)	*r* = 0.470 *p* < 0.001	*r* = −0.203 *p* < 0.001	*r* = −0.282 *p* < 0.001	*r* = −0.620 *p* < 0.001	*r* = −0.139 *p* < 0.001
Vitality (*n* = 892)	*r* = 0.152 *p* < 0.001	*r* = −0.198 *p* < 0.001	*r* = −0.139 *p* < 0.001	*r* = −0.315 *p* < 0.001	*r* = −0.088 *p* = 0.009
Role Functioning (*n* = 898)	*r* = 0.334 *p* < 0.001	*r* = −0.305 *p* < 0.001	*r* = −0.339 *p* < 0.001	*r* = −0.513 *p* < 0.001	*r* = −0.163 *p* < 0.001
Health Perceptions (*n* = 891)	*r* = 0.256 *p* < 0.001	*r* = −0.263 *p* < 0.001	*r* = −0.198 *p* < 0.001	*r* = −0.417 *p* < 0.001	*r* = −0.215 *p* < 0.001
Emotional Functioning (*n* = 889)	*r* = 0.107 *p* = 0.003	*r* = −0.176 *p* < 0.001	*r* = −0.108 *p* = 0.001	*r* = −0.239 *p* < 0.001	*r* = −0.090 *p* = 0.008
Social Functioning (*n* = 878)	*r* = 0.087 *p* = 0.011	*r* = −0.240 *p* < 0.001	*r* = −0.162 *p* < 0.001	*r* = −0.225 *p* < 0.001	*r* = −0.236 *p* < 0.001
Treatment Burden (*n* = 645)	*r* = 0.229 *p* < 0.001	*r* = −0.260 *p* < 0.001	*r* = −0.236 *p* < 0.001	*r* = −0.370 *p* < 0.001	*r* = −0.239 *p* < 0.001
*r* < 0.3 weak correlation		*r* = 0.3–0.49 moderate correlation		*r* > 0.5 strong correlation	

Convergent validity is shown as heat map of *r*-values. *p*-values were assessed by the Spearman’s rank correlation coefficient (*r*). Abbreviations: ppFEV_1_, forced expiratory volume in 1 s (% predicted); MRC, Medical Research Council; QOL-B, Quality of Life Questionnaire-Bronchiectasis.

**Table 7 jcm-11-00441-t007:** MCID estimates for the QOL-B scales.

QOL-B Scale	MCID
Respiratory Symptoms	8.5
Physical Functioning	8.7
Vitality	10.5
Role Functioning	10.4
Health Perceptions	10.3
Emotional Functioning	9.3
Social Functioning	14.1
Treatment Burden	13.0

Abbreviations: MCID, minimal clinical important difference; QOL-B, Quality of Life Questionnaire-Bronchiectasis.

## Data Availability

Data are available from the authors upon request.

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
