# Peer review of "Psychometric Validation of the German Translation of the Quality of Life Questionnaire-Bronchiectasis (QOL-B)—Data from the German Bronchiectasis Registry PROGNOSIS"

_jcm, 2022, doi:10.3390/jcm11020441_

Round 1

Reviewer 1 Report

This is a very nicely written and presented paper that confirms the psychometric properties of the QOL-B in a German registry of patients with bronchiectasis.

I have just two minor points to consider in the discussion

  1. The U.S. FDA prefers anchor-based methods to calculated MCID - is that what you meant by "patient-derived"?  Could you clarify and comment on whether those could be more relevant vs. statistical methods such as the one you present?
  2. Table 6 is the only data that raises some question about the QOL-B domains, with weak correlation between most measures of health status.  One reason could be that they are not measures of current symptom burden; do you have any other thoughts about why that might be to incorporate into the discussion?

Reviewer 2 Report

Dear Editor,

Thank you for asking me to review the manuscript entitled “Psychometric Validation of the German Translation of the Quality of Life Questionnaire-Bronchiectasis (QOL-B) – Data from the German Bronchiectasis Registry PROGNOSIS”. This is a well written manuscript that presents data for the translated QOL-B from 904 patients with bronchiectasis. This is a well-needed study, as accessing the validity of a questionnaire is a continuous process and it provides new insights on how the questionnaire responds in the German population.

The main limitation of the study is that there is no reference to a robust backward and forward translation process for the QOL-B and cultural adaptation through patient interviews. This should be acknowledged, despite appearing to perform similarly to the original English version. It is a shame that the registry’s annual follow up could not provide an opportunity to assess responsiveness, but this has been recognised by the authors.

The authors need to correct a few phrases, as their wording and interpretation of data does not always accurately reflect the stated values. For instance, labelling a correlation of r = -0.513 as strong seems unfair. Please correct your interpretation in lines 258-265 and amend the corresponding methods section, table 6 heat map and discussion.

Finally, I suggest adding a clear column about missing data on Table 2. Patients not receiving bronchiectasis treatment were instructed to skip the Treatment Burden scale and this is clear in the methods but the fact that one out of eight scales of the QOL-B was not relevant to 259 patients needs to be part of the discussion.

Other comments:

Page 1

Line 32: “We observed no relevant floor or ceiling effects”. According to Table 2, this is incorrect. Replace with “We observed relatively small floor and ceiling effects, not exceeding 10.9%.”

Line 33: Cronbach’s alpha 0.73 is not excellent, please replace.

Line 34: Replace “all” with “most”.

Line 38: Replace “and Respiratory Symptoms, Physical Functioning, Vitality, Role Functioning, Health Perceptions as well Treatment Burden” with “and QOL-B, except Social Functioning and Emotional Functioning”.

Line 41: Delete “for the Respiratory Symptoms and Social Functioning scale, respectively”. You do not need this information, you report the whole questionnaire previously.

Line 42: Replace “proved to be a valid and reliable tool to assess QOL among” with “has good validity and test-retest reliability in”.

Page 2

Line 70: Delete “its broader use in clinical practice is still lacking and” as this is not something that you assessed.

Line 73: Replace “determine disease-specific” with “investigate the”.

Line 76: Delete “excellent”.

Line 137: Add the scales’ score range.

How was average daily sputum volume measured?

Page 4

Line 156: “Rho (r) values <0.3 indicate weak, values of 0.3 to 0.49 moderate and values >0.5 strong correlation.” Could you use a cut off system that allows for a more accurate interpretation? I would never consider a correlation strong if it was 0.53.

Line 192: Delete “relevant”.

Line 195: Add ceiling effect data in role functioning and amend the presented threshold for the remaining scales.

Page 7

Line 236: Delete “according to functional categories”.

Page 9

Line 276: Delete “with an average of 10.6 points over all scales”, as there is no meaning in calculating this.

Line 283: Replace “robust” with “valid”.

Line 283: Delete “and is feasible for application in a real-world setting”. The study did not assess feasibility.

Line 280: Delete “respective Spanish translation” with “QOL-B version 3.0”.

Line 290: Replace “no” with “small”.

Line 295: Replace “excellent” with “high to excellent” internal consistency.

Line 302: Delete “good”.

Line 305: What was the MCID range in the Spanish study for the remaining scales?

Page 10

Line 332: Delete “discrete”.

Line 333: “Data on its responsiveness from RCTs are conflicting”. Specify that only Respiratory symptoms is accounted for or include all the QOL-B scale results.

Line 343: Delete “becoming”.

Line 338: Delete “indicating that responsiveness may largely depend on differences in patient populations and the type of intervention [14,16]” as this is not accurate.

Line 366: Delete “meaningful and”.

Line 382: “QOL is not only an important PROM in RCTs but also matters to patients in their everyday lives”. Add reference.

383: Replace “robust” with “valid”.

384: Delete “use and”, as you provide evidence for interpretation only.

Line 388: Delete “and demonstrated its feasibility to determine disease-specific QOL”.

References:

Replace abstracts with the full papers published by Spinou et al., Aksamit et al, and De Souza et al, and any other you identify.
